# City-Scale Mapping of Urban Façade Color Using Street-View Imagery

Teng Zhong [1,2,3], Cheng Ye [1,2,3], Zian Wang [1,2,3], Guoan Tang [1,2,3], Wei Zhang [4,5,*] and Yu Ye [6,7]

1   Key Laboratory of Virtual Geographic Environment (Ministry of Education), Nanjing Normal University, Nanjing 210023, China; tzhong27@njnu.edu.cn (T.Z.); yecheng0728@njnu.edu.cn (C.Y.); wangzian@njnu.edu.cn (Z.W.); tangguoan@njnu.edu.cn (G.T.)
2   State Key Laboratory Cultivation Base of Geographical Environment Evolution, Nanjing Normal University, Nanjing 210023, China
3   Jiangsu Center for Collaborative Innovation in Geographical Information Resource Development and Application, Nanjing Normal University, Nanjing 210023, China
4   Key Laboratory of Metallogenic Prediction of Nonferrous Metals and Geological Environment Monitoring (Ministry of Education), School of Geosciences and Info-Physics, Central South University, Changsha 410083, China
5   School of Municipal and Mapping Engineering, Hunan City University, Yiyang 413000, China
6   College of Architecture and Urban Planning, Tongji University, Shanghai 200092, China; yye@tongji.edu.cn
7   Key Laboratory of Ecology and Energy-Saving Study of Dense Habitat, Tongji University, Shanghai 200092, China
*   Correspondence: viviengis@csu.edu.cn

**Abstract:** Precise urban façade color is the foundation of urban color planning. Nevertheless, existing research on urban colors usually relies on manual sampling due to technical limitations, which brings challenges for evaluating urban façade color with the co-existence of city-scale and fine-grained resolution. In this study, we propose a deep learning-based approach for mapping the urban façade color using street-view imagery. The dominant color of the urban façade (DCUF) is adopted as an indicator to describe the urban façade color. A case study in Shenzhen was conducted to measure the urban façade color using Baidu Street View (BSV) panoramas, with city-scale mapping of the urban façade color in both irregular geographical units and regular grids. Shenzhen's urban façade color has a gray tone with low chroma. The results demonstrate that the proposed method has a high level of accuracy for the extraction of the urban façade color. In short, this study contributes to the development of urban color planning by efficiently analyzing the urban façade color with higher levels of validity across city-scale areas. Insights into the mapping of the urban façade color from the humanistic perspective could facilitate higher quality urban space planning and design.

**Keywords:** urban color planning; urban façade color; street-view images; deep learning; Shenzhen

## 1. Introduction

With the increasing demand for urban sustainability and quality-of-life, it has become important to understand the relationship between urban color and environmental, recreational, and aesthetic conditions within urban areas from a humanistic perspective [1,2]. Modeling urban color is a major urban design element when planning urban spaces [3,4]. Urban façade color is a fundamental component in the cityscape, providing urban residents with unique visual experiences and spatial environment perception, which affect residents' emotions and behaviors [5,6].

Urban façade color planning has been a common practice in European cities since the late 1960s [7–9] and China has focused on planning the urban façade color and rooftop color since the year 2000 [5]. Urban color planning does not mean unifying urban buildings into one color, but rather providing guidelines for color use to enable color harmony in

the urban area. When designing new buildings or renovating old buildings in the city, the color environment formed by the surrounding buildings needs to be considered [10].

The color survey to reveal the cityscape color is vital in urban color planning. The general process of urban color planning contains several steps: (1) conduct the color survey in the target area; (2) determine the dominant color of the target area; (3) provide the guideline of color use with recommended and prohibited color chromatograms for the target area [11]. Traditional measurement of the urban façade color is mainly through field surveys, with color charts or spectrophotometric instruments (e.g., a colorimeter or a spectrophotometer). Color charts have been used to measure the urban façade color based on the observer's visual inspection of sampling areas [7,12]; however, measuring the urban façade color with color charts is highly subjective, relying on the observer's personal visual experience. Spectrophotometers and colorimeters are applied to record color measurements on-site, based on results of sampling locations [13,14].

Image-based methods have been developed in recent years to assess the urban façade color, with the rapid development of digital image processing technologies [15,16], and have been implemented by analyzing the urban color characteristics based on the dominant colors of digital images taken from sampling sites [17]. Urban façade color measurements with image-based methods rely heavily on manual image acquisition, which is time-consuming and labor-intensive. Image-based methods are suitable for the assessment of the urban façade color on a small scale; however, when measuring the urban façade color at the city scale, it is difficult to collect a sufficient number of images covering the whole city area, due to limitations imposed by field survey costs.

Street-view imageries, such as Google Street View, Baidu Street View (BSV), and Tencent Street View, are new data sources for quantifying urban physical features and human activities [18,19]. The use of street-view imageries has been applied for investigating the relationship between street greenery and physical activity [20], mapping the spatio-temporal distribution of solar radiation [19], and estimating pedestrian volume [21]. Many studies have focused on accessible green and blue spaces, which are often advocated as significant design elements in the planning of urban spaces [4], but few studies have conducted mapping the urban façade color at the city scale.

There is a trend to incorporate street-view images with deep learning to express and analyze the urban physical environment on a wider scale [22,23]. The success of deep learning and computer vision has provided new opportunities for urban planners and modelers to understand cities with machine eyes [18]. Artificial intelligence technologies have the potential to efficiently identify characteristics from street-view imageries [24]. Semantic image segmentation based on deep learning, such as SegNet [25] and DeepLab V3 [26], has been applied for the automatic recognition of urban elements from massive street-view imageries [27].

Large-scale, fine-grained mapping of the urban façade color from a humanistic per-spective is the foundation of urban color planning. China's urban planning dilemma is that urban color planning can be effectively carried out at the block level but is difficult to implement at the city level. One of the reasons for this is that the traditional measurement of urban façade color is mainly conducted by field surveys with color charts or spectropho-tometric instruments. However, this method is challenging to adapt to the macro-scale urban color planning of Chinese cities. This study's main objective was to develop a quantitative analysis method for mapping the urban façade color at the city scale. The city of Shenzhen, China, was adopted as an example of mapping the urban façade color and the study is expected to provide a feasible deep learning-based method for quantifying the large-scale urban façade color from street-view imageries.

The remainder of the paper is organized as follows. Section 2 introduces the study area and methodology, focusing on the research framework and technical process of quantitatively extracting the DCUF from massive street-view imagery. Section 3 presents the results of urban façade color mapping at the block level and regular grid scale, analyzing the spatial distribution characteristics of Shenzhen's urban façade color at the city scale.

Section 4 discusses the pros and cons of the proposed automatic measurement of the urban façade color at the city scale, supported by deep learning-based algorithms. Section 5 gives the conclusions of this research.

## 2. Materials and Methods

### 2.1. Study Area

This study was conducted in Shenzhen, Guangdong Province, China (Figure 1). Shenzhen is a megacity, with a resident population totaling 13.026 million at the end of 2018, and has nine administrative districts and one new management area, covering a total area of 1997 km$^2$. Shenzhen has always been one of China's pioneer cities for innovative urban planning. Design research has focused on improving the human-centric quality of the built environment [28]. Choosing Shenzhen as a pivotal case on a city-scale mapping of urban façade color using street-view imagery can guide future urban color planning.

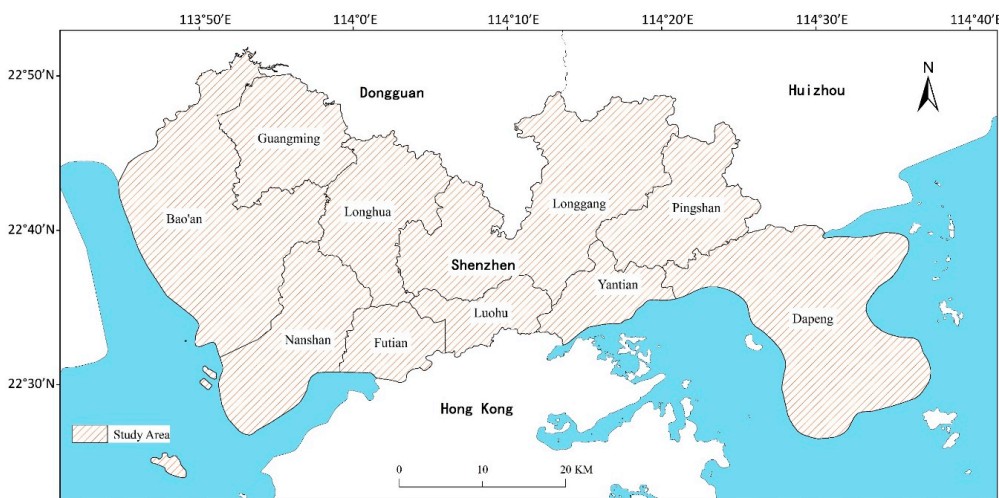

**Figure 1.** Location of case study area: Shenzhen, China.

### 2.2. Dominant Color of the Urban Façade (DCUF)

Using the dominant color as a color descriptor is not a new method. For example, the MPEG standard defined the dominant color descriptor (DCD) as an effective and intuitive quantification factor for expressing image dominant color attributes [29]. The DCD has been widely applied in image classification, image retrieval, and other fields [30,31]. There are some common DCD quantification methods, including the basic histogram-based method and the clustering-based method [32–34].

In this study, the dominant color of the urban façade (DCUF) is proposed as a color descriptor for describing the representative color of the urban façade in an image. The DCUF contains numerous dominant colors in urban façade images. For each dominant color ($C_i$) in the DCUF, the percentage of pixels ($P_i$) in the image area of the urban façade of the color cluster corresponding to $C_i$ is used to describe the weighting of the dominant color ($C_i$). The *DCUF* of an image can be defined with Equation (1):

$$DCUF(I) = \{\{C_i, P_i\}, i = 1, \ldots N\}, \tag{1}$$

where *N* refers to the number of dominant colors in the urban façade image *I*.

There are thousands of color values in an image; therefore, the extraction of the DCUF needs to divide the set of pixel values into several color clusters. Each pixel's color information in the urban façade's image area can be expressed as a point in the hue, saturation, value (HSV) three-dimensional color space. The dominant color of the image can be extracted through cluster analysis of the scattered points in the HSV color space. For the simplicity and feasibility, this study adopted K-means clustering to identify the color clusters for the extraction of the DCUF [32,33]. K-means clustering is one of the

most common methods to partition the data space into clusters. K-means is a method that iteratively divides observations into K categories based on minimizing the error function. In this study, we deliberately set the number of color clusters to be four, as previous studies have found four clusters are significant enough in typological development, and correspond to people's color abstraction of visual space [16].

The results of the K-means clustering of the colors of the urban façade images need to be matched to the color classification standard. The Chinese Building Color Card (CBCC) is the combination of 1026 colors that are defined in the China National Standard GB/T18922-2008 on methods of color specification for architecture. In practice, the CBCC-240 is intensively used in urban color planning, as it is more convenient and practical than the CBCC. The Chinese Building Color Card 240 (CBCC-240) selects 240 commonly used architectural colors from the 1026 colors in the CBCC. In this study, the CBCC-240 is used as the color classification standard for color matching.

The Euclidean distance of HSV was applied to match the results of the K-means clustering colors of the urban façade images to the 240 standard architectural colors defined in CBCC-240. The implementation steps are as follows:

1.  As the CCBC-240 uses the Munsell color system to represent the color, the Munsell color codes of the 240 standard colors are converted to the HSV color space to obtain the corresponding HSV values of the 240 standard colors.
2.  The K-means clustering of the colors of the urban façade images is matched to the 240 standard colors in CBCC-240, using the smallest Euclidean distance (*d*) between the HSV values. The Euclidean distance between the HSV value of a K-means clustering color and the HSV value of the 240 standard colors in the CBCC-240 is calculated using Equation (2) [35]:

$$d = \|(vscos(2\pi h), vssin(2\pi h), v) - (v's'cos(2\pi h'), v's'sin(2\pi h'), v')\|, \tag{2}$$

where (*h*, *s*, *v*) is the HSV value of the K-means clustering color, and (*h'*, *s'*, *v'*) is that of the CBCC-240 sample colors. The value range of each color channel is normalized to [0, 1].

Figure 2 shows a flowchart to illustrate the extraction of the dominant color of the urban façade (DCUF) from the street view image at a location. This flowchart has three parts, including (1) Street-view imagery acquisition; (2) Building façade extraction via Deeplab V3+ model; (3) Dominant color extraction from building façade. Details of the implementation are explained in the following sections.

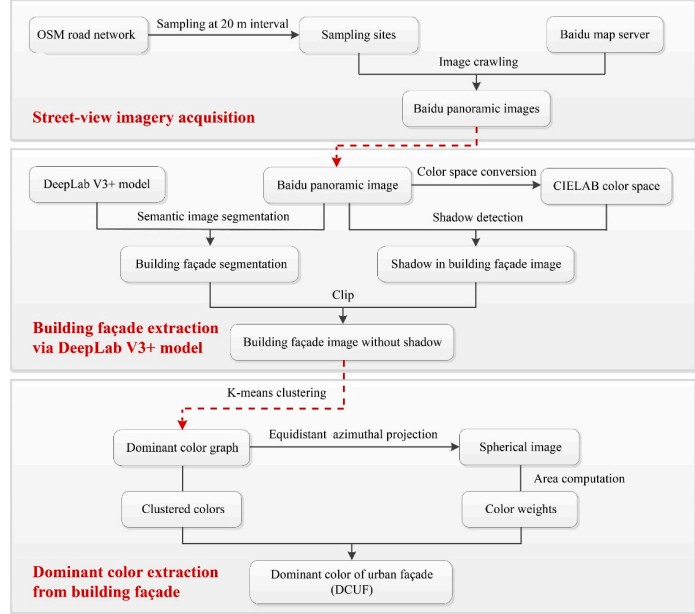

**Figure 2.** Flowchart of determining the DCUF from the street view image at a location.

### 2.3. Baidu Street View (BSV) Panorama Acquisition

The BSV panoramas were automatically crawled based on the locations of the sampling points and the Baidu Map Application Programming Interface (API) service. Open-StreetMap (OSM) is a collaborative project to create and provide free geographic data. The road network in the study area was derived from OSM (https://www.openstreetmap.org/, accessed on 11 April 2021) with Python scripts. The sample locations were generated at intervals of 20 m. A total of 231,758 sample points were generated based on the simplified road network from OSM. Figure 3 presents the illustration of the sampling points on the road network, with a randomly selected area that has been magnified. The latitude and longitude coordinates of the sampling points were used as the major parameters for crawling the BSV panoramas via the Baidu Map API.

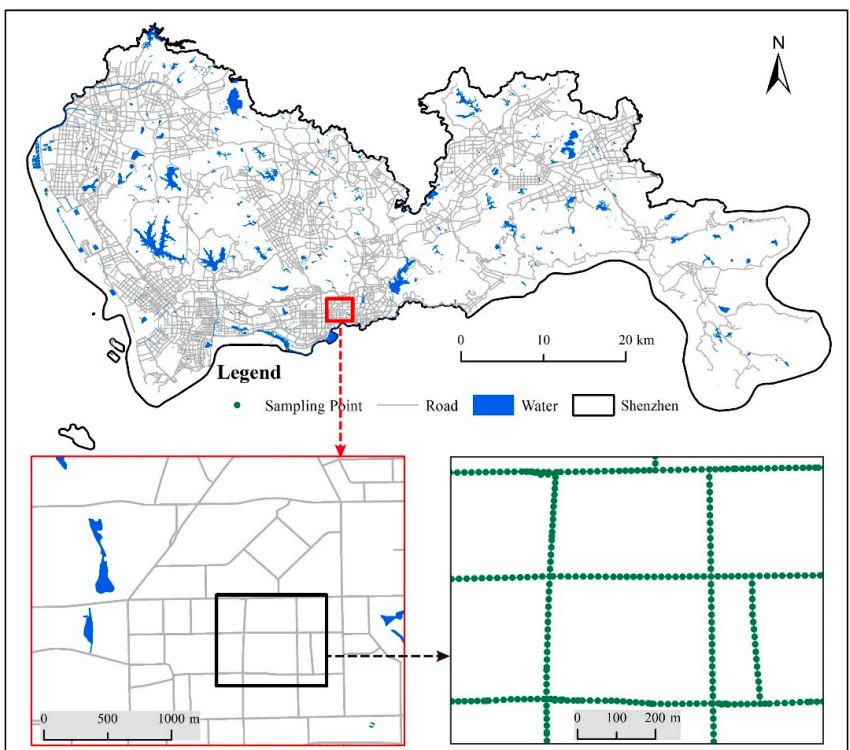

**Figure 3.** Sampling points collected across the road network of Shenzhen.

The BSV panoramas were captured by cameras mounted on Baidu street-view vehicles from a height of 2.5 m. The Baidu Maps Open Platform API provides a service for street-view image query and this was downloaded. By defining URL parameters sent through a standard HTTP request, users can access and obtain a panoramic image of a specific image's collection point [27]. When making an HTTP request, it is necessary to provide the location, Field of View (FOV), heading, pitch, and other information for image retrieval. In this experiment, the parameters are unified as: image size 1024 × 512, FOV = 360°.

Where the APIKEY is the user's access key, WIDTH and HEIGHT are the width and height of the street-view image, LAT and LON are the latitude and longitude coordinates of the sampling point, and FOV is the horizontal range. In this study, the image size of the BSV panorama was set to 1024 × 512, and the field of view (FOV) was set to 360°.

The image quality of the BSV panoramas varies with environmental factors such as the atmosphere and climate, which affect the accuracy of semantic image segmentation and further impact the color feature extraction [36]. Therefore, the original BSV panoramas need to be pre-processed to refine the quality of the images. The images were detected to find out the ones with color shift and abnormal brightness. Then, white balance algorithm and Gamma correction were used to correct the color shift and brightness, respectively.

*2.4. Extraction of the Urban Façade from BSV*

The image area of the urban façade needs to be extracted from the BSV panoramas. In this study, the semantic image segmentation based on the DeepLab V3+ model was used to extract the urban façade. Next, the image shadow detection method based on the CIELAB color space was used to detect shadow areas [36–38]. The identified shadow area was removed from the image of the urban façade to mitigate the effect of shadow on the extraction of the DCUF.

2.4.1. Deep Learning-Based Extraction of the Urban Façade

DeepLab V3+ was used to extract the urban façades from the street-view images automatically. DeepLab is a series of open-source semantic image segmentation models based on the Convolutional Neural Network (CNN) using TensorFlow, which can be applied for image pixel-level classification [26]. The Cityscapes Dataset is a benchmark suite and large-scale dataset for semantic urban scene understanding, including a comprehensive range of complex urban street scenes from 50 different cities [39]. The DeepLab V3+ model trained with Cityscapes has been proven reliable in semantic image segmentation applications [40,41]. This study used the Cityscapes dataset to train the semantic image segmentation model to extract the urban façade.

Figure 4 is an illustration of the urban façade extraction from the BSV panorama. The semantic image segmentation model divides the image into several segmentations (Figure 4b). The urban façade image is then extracted by using the segmented urban façade boundary to clip the street-view image (Figure 4c). To further improve the urban façade extraction results, both closing and erosion algorithms were adopted to refine the building's edge by removing the fragmented areas [42].

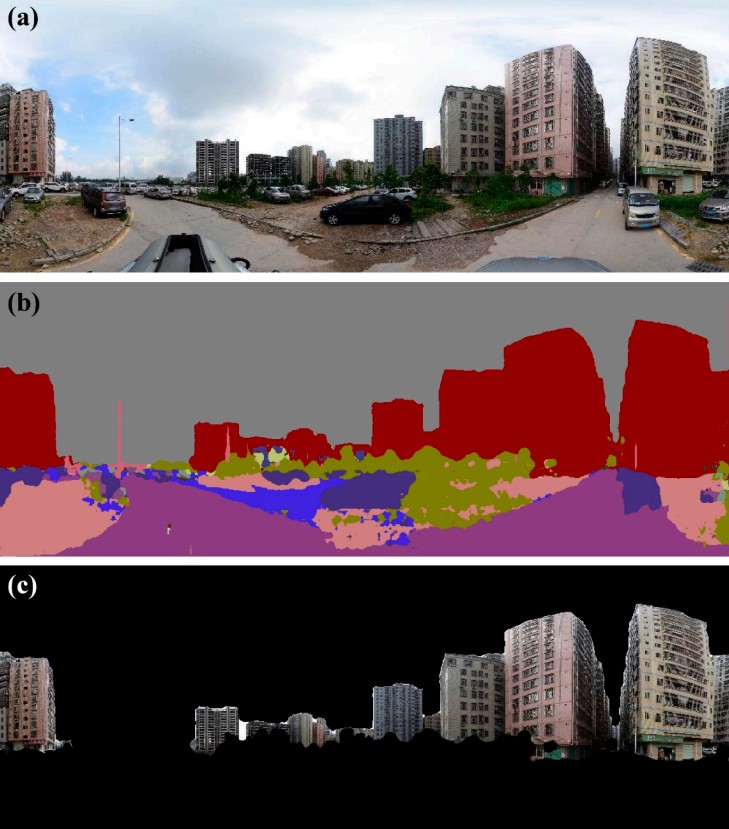

**Figure 4.** An illustration of the urban façade extraction from street-view imagery with the deep learning-based method. (**a**) An example of a BSV panorama in the study area, (**b**) results of semantic image segmentation with red polygons indicating the image area of the urban façade, (**c**) results of the extracted urban façade image.

### 2.4.2. Shadow Detection of the Urban Façade Using CIELAB Color Space

Shadow areas will appear on the image if the incident light source is blocked by an opaque object when the imaging device collects the image. Shadows cause uneven brightness and darkness of urban façade images. The distortion of image color information will ultimately lead to errors in the extraction of dominant colors. An image shadow detection method was adopted based on the CIELAB color space to eliminate the influence of shadows on the extraction results of DCUF [38,43]. There are three color channels in the CIELAB color space, namely *L*, *A*, and *B*. Channel L represents brightness, and the channel *L* values are in the interval of 0 to 100. Channel *A* and Channel *B* are the other two color channels in the CIELAB color space, with values in the range of 128 to 127. As the shadow is darker than the surrounding environment, the pixel's channel *L* value in the shadow area is significantly lower than that of other pixels, while the channel *B* value is smaller. Setting a threshold considering the combination of *L* and *B* channels can effectively detect the shadow area.

The street-view images were converted from HSV color space to CIELAB color space. The *L*, *A*, and *B* values are normalized to [0, 255] to eliminate the effect of inconsistency. The shadows were detected according to Equation (3) [43]:

$$
S = \begin{cases} 1, & \left( \begin{array}{l} mean(A) + mean(B) \leq 256, L \leq meanL - std(L)/3 \\ or \quad mean(A) + mean(B) > 256, mean(L) + mean(B) \leq 200 \end{array} \right) \\ 0, & \left( \begin{array}{l} mean(A) + mean(B) \leq 256, L > mean(L) - std(L)/3 \\ or \quad mean(A) + mean(B) > 256, mean(L) + mean(B) > 200 \end{array} \right) \end{cases} , \quad (3)
$$

where *S* represents the shadow detection result, with 1 for a shadow pixel, and 0 for a non-shadow pixel. The *mean(L)*, *mean(A)*, and *mean(B)* values represent the average value of all pixels in a single image in the three channels of *L*, *A*, and *B*, respectively. The *std(L)* value represents all pixels' standard deviations in a single image in the *L* channel.

The shadow detection result of a street-view image is shown in Figure 5. Figure 5b is the shadow detection result on the extracted urban façade image. The detected shadow area will be removed from the extracted urban façade image to eliminate the effect of shadows on the extraction of the DCUF.

**(a)**

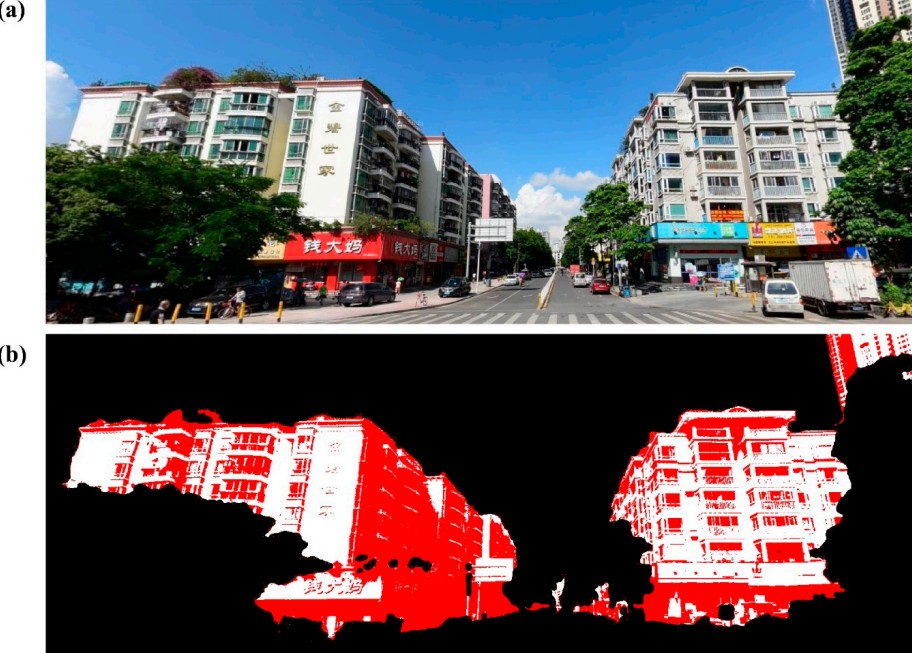

**(b)**

**Figure 5.** An illustration of shadow detection from a street-view image. (**a**) An example of BSV panorama with shadow, (**b**) the shadow detection result on the extracted urban façade image.

The error of the results of dominant color evaluation increase when the proportion of shadow areas is changed. At present, to eliminate the effect of the shadow area on the extraction of DCUF from street-view images, we chose to delete pixels of the building facade in the shadow area and use those image pixels of the building façade not in the shadow area to extract DCUF. Under some specific circumstances, the street view images collected at some sampling points are not suitable for evaluating urban façade color. In this study, when the number of pixels of the building façade not in the shadow area is less than 5000, the evaluation is stopped. The threshold value 5000 is determined arbitrarily according to the opinions of the experts in urban planning. Future work is needed to develop a more sophisticated algorithm to determine the threshold value.

### 2.5. Determination of the Weight of the Dominant Color in the DCUF

In addition to the colors obtained by clustering, each color's weight needs to be calculated to construct the DCUF. The commonly used method is to determine the color weight by calculating each color's pixel area's proportion. From a human-oriented perspective, the human perception of natural scenes is based on a spherical surface rather than a plane. The original BSV panorama downloaded via the Baidu Maps Open Platform API is in the form of an equidistant cylindrical projection, and is a plane projection (Figure 6a). Therefore, BSV panoramas need to be transformed from plane projection to spherical projection. BSV panoramas were converted from an equidistant cylindrical projection to an equidistant azimuthal projection to create a spherical image [19].

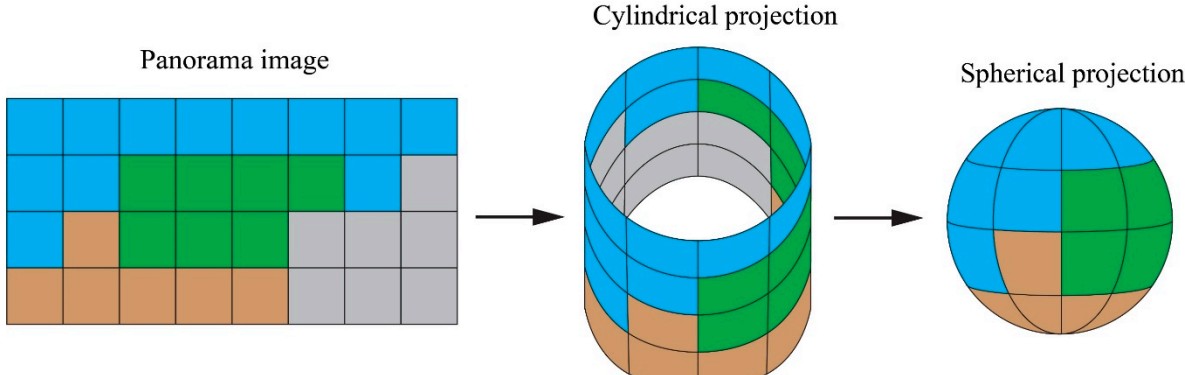

### (a) Transformation of panorama image into spherical projection

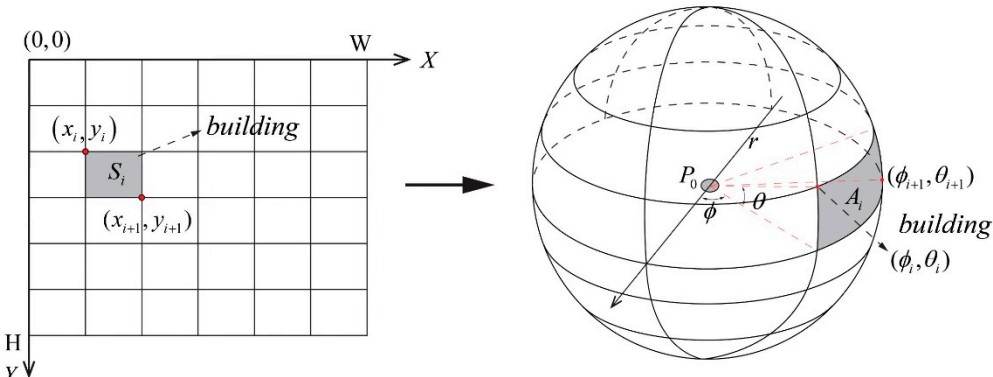

### (b) Schematic of projection calculation

**Figure 6.** The geometrical transform of equidistant cylindrical projections. (**a**) An illustration of the transformation of a panorama image into a spherical projection, (**b**) schematic of the transformation from an equidistant cylindrical projection to an equidistant azimuthal projection.

Figure 6b illustrates the geometric model of transforming equidistant cylindrical projections into equidistant azimuthal projections. Each coordinate on the result spherical image can be calculated with Equation (4):

$$\begin{cases} \theta_i = \dfrac{0.5H - y_i}{r} = \dfrac{\pi H - 2\pi y_i}{W} \\ \phi_i = \dfrac{x_i}{r} = \dfrac{2\pi x_i}{W} \end{cases}, \tag{4}$$

where $(x_i, y_i)$ is the corresponding coordinate of $(\phi_i, \theta_i)$ on the cylindrical panorama; $H$ and $W$ are the height and width of a cylindrical panorama, respectively; $r$ is the radius of the spherical image, which should be $r = W/2\pi$.

Then, each pixel area $A_i$ and solid angle $\Omega_i$ in a spherical image can be calculated with the Equations (5) and (6) [19,44]:

$$A_i = r^2 \int_{\phi_i}^{\phi_{i+1}} d\phi \int_{\theta_i}^{\theta_{i+1}} \cos\theta d\theta \tag{5}$$

$$\Omega_i = \frac{A_i}{r^2} = \int_{\phi_i}^{\phi_{i+1}} d\phi \int_{\theta_i}^{\theta_{i+1}} \cos\theta d\theta, \tag{6}$$

where $(\phi_i, \theta_i)$ is the coordinate on the cylindrical panorama.

All pixels of the same color were counted based on the spherical image to determine the weight of the dominant color in the DCUF. The color weight is arranged in descending order to realize the extraction of the DCUF, and construct the DCUF database.

The extraction results of the DCUF at three sampling sites are shown in Figure 7. For example, as shown in Figure 7a, four dominant colors, $c_1$, $c_2$, $c_3$, and $c_4$, were extracted from the BSV panorama collected at the sampling location. The corresponding weights of the dominant colors $c_1$, $c_2$, $c_3$, and $c_4$ in the DCUF were 0.32, 0.26, 0.25, and 0.17, respectively.

**(a)**

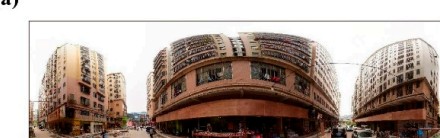

| Color | Weight | HSV Code | Munsell Color Code |
|---|---|---|---|
| $c_1$ | 0.35 | (21, 0.22, 0.69) | 6.9YR 6.5/2 |
| $c_2$ | 0.31 | (11, 0.43, 0.62) | 0.6YR 5/4.8 |
| $c_3$ | 0.31 | (7, 0.03, 0.93) | 9.4RP 9/1 |
| $c_4$ | 0.03 | (30, 0.01, 0.94) | 7.5GY 9/1 |

**(b)**

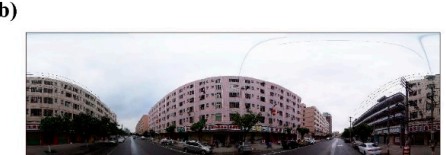

| Color | Weight | HSV Code | Munsell Color Code |
|---|---|---|---|
| $c_1$ | 0.37 | (333, 0.07, 0.51) | 2.5RP 5.5/1 |
| $c_2$ | 0.34 | (351, 0.04, 0.67) | 3.8RP 7/1 |
| $c_3$ | 0.23 | (8, 0.18, 0.51) | 8.8R 5/1.6 |
| $c_4$ | 0.06 | (210, 0.07, 0.92) | 7.5PB 9/1.6 |

**(c)**

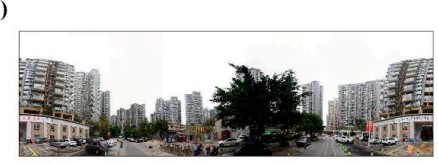

| Color | Weight | HSV Code | Munsell Color Code |
|---|---|---|---|
| $c_1$ | 0.32 | (82, 0.06, 0.75) | 7.5GY 7.5/1 |
| $c_2$ | 0.30 | (30, 0.01, 0.94) | 7.5GY 9/1 |
| $c_3$ | 0.25 | (202, 0.06, 0.74) | 10B 7.5/1 |
| $c_4$ | 0.13 | (200, 0.03, 0.91) | 1.3P 9/1 |

**Figure 7.** The extraction result of DCUF of three sample BSV panoramas. (**a**) An illustration of the BSV panorama and the corresponding four dominant colors in the DCUF, (**b**) the extraction results of four dominant colors in the DCUF at another sampling site, (**c**) the illustration of the four dominant colors in the DCUF at the third sampling site.

The DCUF data structure was developed to manage the DCUF extracted from massive street-view imageries efficiently. The DCUF data structure is shown in Table 1. "ID" is the unique identifier of the DCUF extracted at the sampling location. The geographic coordinates of the street-view image sampling points are stored in the field of "Location". The HSV values of the four dominant colors and the corresponding color weights and Munsell color codes are recorded in the field of "Dominant Color" in descending order according to the color weights.

**Table 1.** The data structure of the DCUF. The Munsell color system divides hues into ten main colors: red (R), yellow–red (YR), yellow (Y), green–yellow (GY), green (G), blue–green (BG), blue (B), purple–blue (PB), purple (P), and red–purple (RP).

| ID | Location | Dominant Colors |
|----|----------|-----------------|
| 1 | (113.9195, 22.5128) | (0.35, (21,0.22,0.69),"6.9 YR 6.5/2"), (0.31, (11, 0.43, 0.62), "0.6 YR 5/4.8"), (0.31, (7, 0.03, 0.93), "9.4 RP 9/1"), (0.03, (30, 0.01, 0.94), "7.5 GY 9/1") |
| 2 | (113.9012, 22.8101) | (0.37, (333, 0.07, 0.51),"2.5 RP 5.5/1"), (0.34, (351, 0.04, 0.67), "3.8 RP 7/1"), (0.23, (8, 0.18, 0.51), "8.8 R 5/1.6"), (0.06, (210, 0.07, 0.92), "7.5 PB 9/1.6") |
| 3 | (113.8489,22.5709) | (0.32, (82, 0.06, 0.75),"7.5 GY 7.5/1"), (0.30, (30, 0.01, 0.94), "7.5 GY 9/1"), (0.25, (202, 0.06, 0.74), "10 B 7.5/1"), (0.13, (200, 0.03, 0.91), "1.3 P 9/1") |

## 3. Results

The extraction result of the DCUF at each sampling point was applied to map the spatial distribution of urban dominant colors in Shenzhen. The DCUF at each sampling point has four dominant colors with corresponding color weights, indicating the proportions in the image area of the urban façade. To map the city-scale urban façade color, we adopted the dominant color with the largest color weight in the DCUF to represent the urban façade color at each sampling point. The mapping of urban façade colors in Shenzhen was conducted based on both irregular geographical units and regular grids.

### 3.1. Mapping of Urban Façade Color Based on Street and City Block

Figure 8 is the spatial distribution of the urban façade color in Shenzhen based on the street and city block. In the study, a total of 207,401 BSV panoramas were collected from 231,758 sampling points. The reason for this mismatch between the number of valid BSV panoramas and sampling points is that BSV panoramas do not cover all the road networks in Shenzhen. The locations of the sampling points are determined according to the road network from OSM. There was no BSV panorama in the road segments of some remote areas. Figure 8a shows the spatial distribution of the urban façade color of each BSV sampling point with a valid BSV panorama.

The urban façade color can be aggregated into different streets. The urban façade color on each street segment was determined by overlaying the urban façade color of BSV sampling points on the street segments (Figure 8b). The street segment's urban façade color was calculated based on the majority value of the urban façade color of BSV sampling points located on the street segment. We further generated the urban façade color mapping result based on the city blocks (Figure 8c). The majority value of the urban façade color of BSV sampling points located on the city block represents the city block's urban façade color.

Overall, the urban façade color of Shenzhen is characterized by a gray tone with low chroma. As shown in Figure 8, the urban façade color with the largest proportion of each BSV sampling point is medium-gray (Munsell code: 10B 5.25/1), while that of each street and city block is a dark-gray color (Munsell code: 5PB 3.5/1). Such light and cool colors of urban façades are in harmony with the natural environment of the coastal city. In Nanshan, Futian, Luohu, and Yantian District, the urban façade color is medium-gray, dark-gray, and blue–gray, respectively, because vast commercial buildings with substantial curtain walls of glass exist in these areas, leading to a cool color palette. The urban façade color is more diverse in other districts, such as Dapeng, Guangming, and Pingshan District. The urban façade colors here are light-gray, brown–gray, and some light and warm colors in residential areas, such as brown, light yellow, and cream-colored.

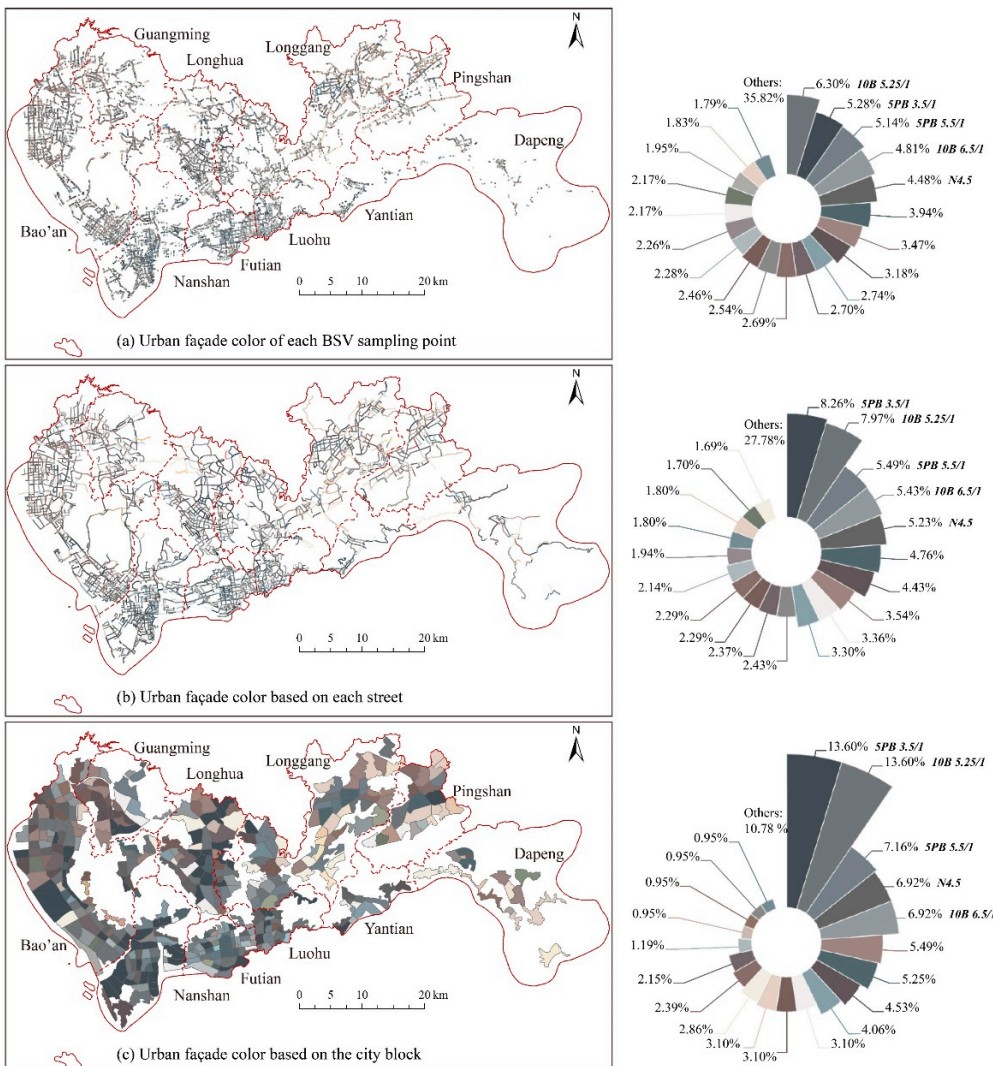

**Figure 8.** The mapping of the urban façade color in Shenzhen based on geographic units. (**a**) Urban façade color of each BSV sampling point, (**b**) urban façade color based on each street, (**c**) urban façade color based on the city block.

### 3.2. Mapping of the Urban Façade Color Based on a Regular Grid

The mapping of the urban façade color in a regular grid of different grid sizes can facilitate the comparison of the urban façade color at different scales. This study used several grid units, including 500 m, 1000 m, and 2000 m, to explore the spatial variation of the urban façade color with a multi-scale comparative analysis (Figure 9). The gird's urban façade color was calculated based on the majority value of the urban façade color of BSV sampling points located on the grid.

As shown in Figure 9, the dark-gray color (Munsell code: 5PB 3.5/1) is the urban façade color with the largest proportion in all three grid sizes. In the grid-scale of 500 m (Figure 9a), the urban façade color of Shenzhen is a gray tone, together with some bright and high chroma tones such as red, blue, and green. The bright colors are the various façade colors of some hotels or commercial buildings. Some commercial buildings have adopted intense and jumping colors, which are in sharp contrast with office buildings. As the grid size increases, the detailed color information in small unit sizes is integrated.

At the grid-scale of 2000 m (Figure 9c), the dark-gray color is distributed in patches, and mainly distributed in Nanshan District, and Longhua District. Brown and brown–gray colors are concentrated in Longgang District, Pingshan District, and Dapeng District, but rarely appear in Nanshan District, Futian District, and Luohu District. Light yellow is

occasionally distributed in Longgang District, Pingshan District, and Longhua District. Overall, the colors are more diverse and brighter in Longgang District, Pingshan District, and Dapeng District than in Nanshan District, Futian District, and Luohu District.

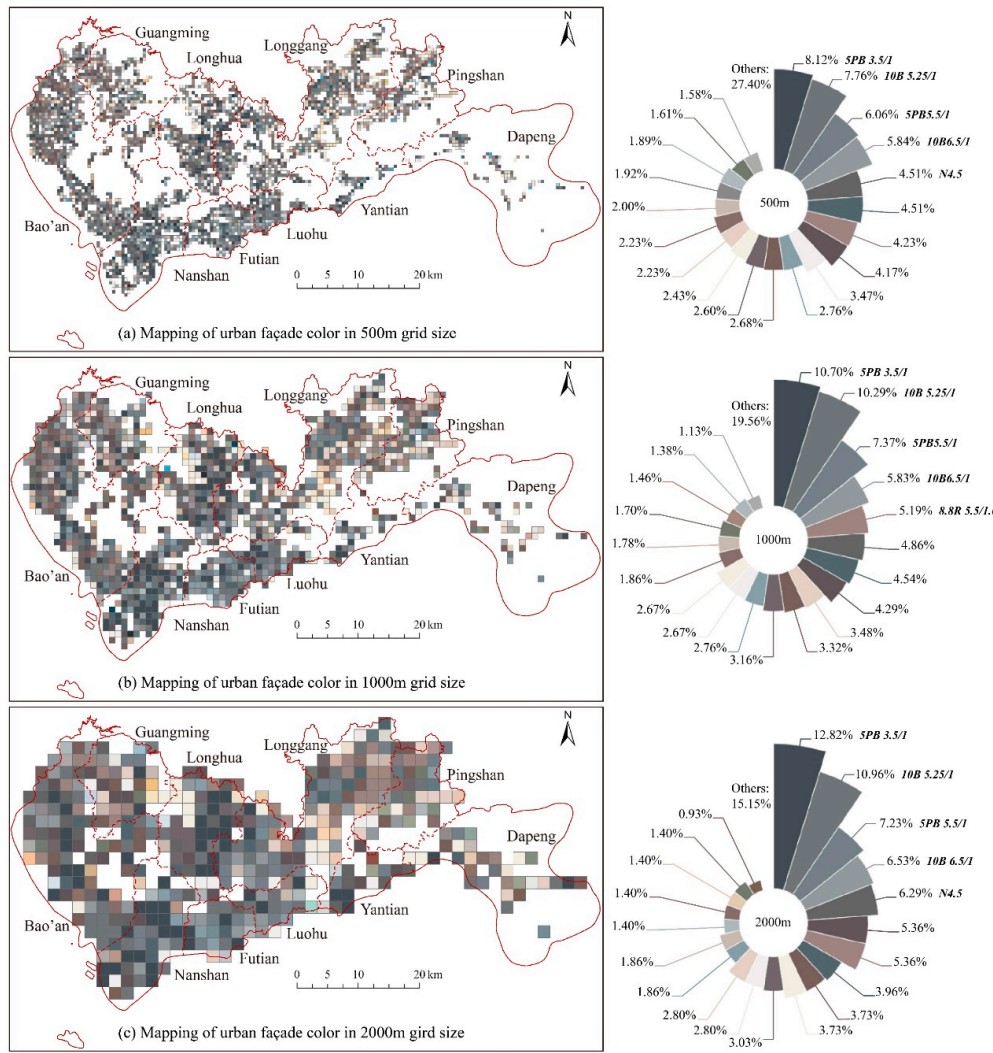

**Figure 9.** Comparison of mapping urban façade color of Shenzhen in different grid size.

### 3.3. Result Verification

The performance of the extraction method of the urban façade color was evaluated with the expert scoring method. A C# software for result verification was developed for experts to determine whether the extracted four dominant colors with the largest DCUF weight represent the dominant color of urban façade based on the BSV panorama of the site. Figure 10 shows the user interface of the C# software to collect feedback from experts.

Urban color planning is a relatively professional field in urban planning and design. In the past, investigators in urban color surveys have mainly trained professionals to reduce the influence of subjective factors on the color survey results. In this study, we invited 30 urban planning and design experts to verify our experimental results of the automatic extraction of DCUF from the street view images [22].

To ensure that samples for verification are evenly distributed with randomness in the test area, this study divides the study area into grids of 1000 m in size. There are a total of 1234 grids with valid data of street view images. The dataset for verification is constructed by selecting one street view image from each grid. We invited 30 experts with expertise in urban planning and design to evaluate the extracted urban façade color based on the four dominant colors with the largest weight in the DCUF. The verification results show that the

accuracy of our DCUF extraction results is 83.6%, and the standard deviation is 0.07, which means the experts think the extracted DCUFs have a high success rate in representing the dominant color of the urban façade.

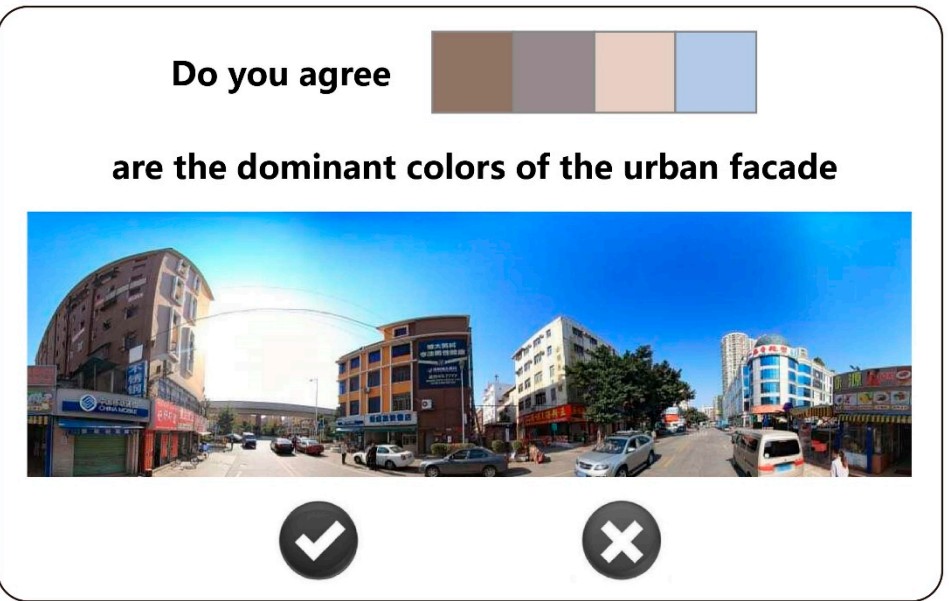

**Figure 10.** The user interface of the software developed for results verification.

## 4. Discussion

### 4.1. Measuring the Urban Façade Color at the City Scale

Previous research on urban colors has tended to rely on manual sampling due to technology and data limitations. The advantage of the manual sampling analysis method based on small samples is its level of high accuracy. However, implementation of the manual sampling analysis method requires many well-trained investigators, which is challenging for the quantitative evaluation of the urban façade color at the city scale. In response to this problem, this study explores an automated quantitative evaluation method for assessing urban façade color at the city scale, supported by deep learning-based algorithms. Compared with traditional small sample analysis, using massive street-view imagery can provide a border measurement of the urban façade color at the city scale. Using street-view imageries to measure the urban façade color can significantly reduce the labor cost of field surveys, in terms of raw data acquisition.

There is no urban color planning at the city level in Shenzhen. Our research can not only enable the survey on the urban façade color status in Shenzhen at the block level but also help Shenzhen analyze urban façade color status on the city level. Our proposed method can automatically and accurately construct the urban façade color database at the city level. The urban façade color database can describe the urban façade color status, which helps to develop guidelines of color use with recommended and prohibited color chromatograms for the target area.

### 4.2. Contributions for Precise Urban Planning and Design

The construction of the urban façade color database by measuring the dominant colors of urban façades in different city areas is a major component of urban planning and design. The proposed method for measuring urban façade color from the humanistic perspective will contribute to a more accurate evaluation of the urban façade color, which will facilitate the planning and design of higher quality urban spaces. For example, the extracted results of the urban façade color can be used to set a more accurate urban design code for each city area. The urban design code will provide strong support for small-scale urban design and color selection for micro-updates. Considering that the street view data

provided by the map service provider will be continuously updated, in the future, we could develop an interactive management system of the urban façade color at the city scale. The management system based on the updated street-view imageries can be used to perform real-time analysis of urban façade colors, which can also help achieve the effective management and real-time monitoring of urban design quality.

### 4.3. Combining Urban Science with Urban Design: A Data-Informed, Algorithm-Driven Perspective

The proposed measurement of the urban façade color is a generic framework that can be easily adopted in practical applications, which serves as the new paradigm for current urban planning and design. This framework uses open access street-view images as the data source, which solves the difficulty of obtaining a basic dataset in actual work. At present, high-precision BSV panoramas have full coverage of China's first- and second-tier cities; in the meantime, BSV panoramas have also covered a considerable number of third- and fourth-tier cities' central areas. Google Street View (GSV) imagery can be used as a data source in countries and regions outside of China. GSV panoramas have achieved high-precision full coverage of most of the major cities around the world.

The proposed deep learning-based algorithm can efficiently analyze the urban façade color with higher validity, which can be used to evaluate and visualize the city-scale urban façade color with high-precision. In addition, we can further improve the accuracy of the deep learning-based measurement of the urban façade color through the training of more samples of street-view imageries.

The variation in urban façade color is related to the social and economic attributes of the city. The color characteristics of the urban facade can reflect the features of the city's functional zoning. For example, due to the prominent office and commercial functions of the Futian District, substantial curtain walls glass of urban facades are blue–gray and medium-grey. In future research, DCUF can be used to construct a quantitative evaluation index of urban color. The DCUF, together with other urban physical environment factors such as land function, green viewing rate, housing price, and further evaluation indexes, construct a city evaluation model to produce social and economic benefits.

In terms of the research orientation of urban color planning, this analytical approach can reveal the distribution characteristics of the urban façade color through quantitative analysis rather than subjective feelings. It will also push the methodology boundary of urban color planning by combining new urban data with new urban analytical techniques.

### 4.4. Limitations and Future Steps

This study has limitations in the data source of street-view imagery. The street-view imageries adopted in this study are BSV panoramas, collected by cameras mounted on BSV cars. Although the BSV panoramas have wide spatial coverage in Shenzhen, these BSV panoramas are collected in different weathers and seasons. We can retrieve the approximate date (e.g., year and month) of when the BSV images were taken; however, the detailed capture time (e.g., day and hour) of BSV is currently unavailable for the time being. It is difficult for us to determine the day and hour that the BSV was collected. Customized street-view image data acquisition is not feasible for us as the collection of street view images is conducted and controlled by online map service providers.

The chroma and brightness of the street-view imageries are affected by different weather patterns and seasons. Collecting street view images in good sky conditions is a benefit for recording more realistic city colors. This research is a preliminary exploration of using street view images to analyze the dominant color of urban façades. In this study, we carried out brightness and contrast adjustments to mitigate the problems. To achieve a more consistent result of the mapping of urban façade color, more cases of the color being affected by weather and season need to be analyzed in order to standardize the color of street-view imageries. In the future, there will be more publicly available street view images, which will enable us to explore the urban façade color with the most suitable street view images.

In this research, pixels of shaded urban façades were directly deleted from images, which causes a loss of area of the urban façades. Future research could use the shadow removal method to remove the effect of shadows on the extraction of the urban façade color from street-view imageries. In addition, in this study, the urban façade color was only analyzed from the perspective of quantitatively measuring the urban physical environment. In the future, there will be a need to further explore the relationship between the urban façade color and residents' mental health.

## 5. Conclusions

This study is an attempt to map the urban façade color using street-view images at the city scale. The DCUF is proposed as a color descriptor for describing the representative color of the urban façade in an image. An automatic method was developed to extract the urban façade color based on the DCUF with a deep learning-based method. We conducted a case study in Shenzhen to measure the urban façade color using BSV panoramas. This study demonstrates that street-view imageries can be used as an important data source for urban color research.

Mapping at different levels revealed that Shenzhen's urban façade color is a gray tone with low chroma. Urban façade colors vary in different districts in Shenzhen. The spatial distribution and quantitative information of the urban façade color could provide a valuable reference for urban color planning in the future. The methodology presented in this study could be applied to quantifying the urban façade color in different cities, given the publicly available street-view imageries.

**Author Contributions:** Conceptualization, T.Z., Y.Y. and G.T.; methodology, T.Z., W.Z. and Y.Y.; experiments, C.Y., W.Z. and Z.W.; data Curation, C.Y. and Z.W.; writing—original draft preparation, T.Z. and W.Z.; writing—review and editing, T.Z., Z.W., G.T. and Y.Y.; funding acquisition, and Y.Y. All authors have read and agreed to the published version of the manuscript.

**Funding:** This research was funded by the National Key R&D Program of China, grant number 2017YFB0503500, the National Natural Science Foundation of China, grant number 52078343, 51878428, and 51708410, and Natural Science Foundation of Shanghai, grant number 20ZR1462200.

**Institutional Review Board Statement:** Not applicable.

**Informed Consent Statement:** Not applicable.

**Data Availability Statement:** Not applicable.

**Acknowledgments:** We would like to thank the editor and anonymous referees for their constructive suggestions and comments that helped to improve the quality of this paper.

**Conflicts of Interest:** The authors declare no conflict of interest.

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
