# Peer review of "City-Scale Mapping of Urban Façade Color Using Street-View Imagery"

_remotesensing, doi:10.3390/rs13081591_

Round 1

Reviewer 1 Report

This manuscript deals with the dominant color mapping for urban streets. The article contains multiple original scientific contributions and matches with the aims and scope of journal of Remote Sensing. However, the aim of the study and the evaluation process were not highlighted well in this manuscript. The following questions are needed to be reviewed before its publication.

  • My major question is where exactly this method can be applied for. Despite of the excellent mapping of urban scale façade color, the discussion on applicability is too weak to highlight the value of the author’s contribution.

  • I recommend the authors to provide another new Figure that presents the detailed process of determining the dominant color at a location from street view image source.

  • Please give some more explanation of the validation method and result.
    The result from few people (10 experts) do not represent the public. The areas of expertise of the "the experts" are not described.
    The validation method seems not reasonable. Only one color is appearing in the questionnaire asking for the agreement. There are four weight ranked colors in Fig. 6. Shouldn't the questionnaire have four or five color selections?
    Maybe, it is better to cite some valid methods used in other studies that similar to the author’s validation method in order to enhance the reliability of validation method and result applied in this study.

  • How did the author control the sky conditions? The façade color and the brightness may be influenced by the sky conditions, rain, snow, illuminance, direct and diffuse component of the solar energy. Please, provide some discussion on this questions. Evenly distributed brightness can be achieved around 12 to 1 PM with overcast sky.

  • The dominant color is determined by counting the number of pixels left. Doesn't the error of the result of dominant color evaluation increase when proportion of shadow areas are changed?
    In case of dense trees standing along the street are detected (or shadow), what is the next step? Shouldn’t it stop evaluation? Or go on to evaluation with the rest of the area?

  • In Section 4, I think it is better to add some discussion on the specific applicability of this method. Using the dominant color of the existing street, what can the user’s do?

  • I think it is better to delete the URL in L 169. And, it may be moved to the reference by citing. The methods to obtain street view image (L171) can also be deleted. I believe the authors can just provide a short information and citation. The domestic URL may not even be opened in some countries.

  • L245, hemispherical image?

Reviewer 2 Report

Thank you for an important contribution in a growing research field. The methods you develop are described in a thorough enough manner. You apply them in a concrete case with results that are comprehensive.

My main concern is the framing of the paper. Why do we need to know the cityscape color? You write that it is important from a humanistic perspective and that it affects residents’ emotions and behaviors.  Then your results are that everything is a range of shades of brown and grey for the whole city.

How come you did not decide to try to instead detect the color of individual buildings? Such information has more applications. It would be more useful for planners and would enable calls for changing of façade color if needed. The methods you use will be used for gathering much more granular information in the years to come. Façade materials, window areas, balconies, down to the number of windowpanes in the windows, will be analyzed using street view images. This data has higher value for authorities, planners, as well as renovation industry. I see your work as one contributing element in this development.

Consider removing section 3.3 Result verification. In my view it adds little to the paper and removes the focus from your contribution.

Reviewer 3 Report

The manuscript presents a method for assessing façade color across an entire city using deep learning and existing data sources. The methods and techniques are thoroughly discussed and well presented. The justification of the mapping is not clear. The authors state that China has planned urban façade color since 2000, and how does this plan compare to the actual results of this study? How would the author's results be used to influence policy? Additionally, does variation in façade color correspond to socio-economic or other variation in the neighborhoods? The authors' methods are engaging with potential applications outside of façade color but would benefit from additional discussion of this research's importance.

Round 2

Reviewer 1 Report

Thank to the authors for the responses.

I have no further questions.